# Predictive Modeling of Suitable Habitats for *Cinnamomum Camphora* (L.) Presl Using Maxent Model under Climate Change in China

**DOI:** 10.3390/ijerph16173185

**Published:** 2019-08-31

**Authors:** Lei Zhang, Zhinong Jing, Zuyao Li, Yang Liu, Shengzuo Fang

**Affiliations:** 1College of Forestry, Nanjing Forestry University, Nanjing 210037, China; 2Co-Innovation Center for Sustainable Forestry in Southern China, Nanjing Forestry University, Nanjing 210037, China; 3College of Water Conservancy and Ecological Engineering, Nanchang Institute of Technology, Nanchang 330099, China; 4College of Forestry, Jiangxi Agricultural University, Nanchang 330045, China

**Keywords:** climate change, *Cinnamomum camphora* (L.) Presl, distribution of habitats, MaxEnt, jackknife test

## Abstract

Rapid changes in global climate exert tremendous pressure on forest ecosystems. *Cinnamomum camphora* (L.) Presl is a multi-functional tree species, and its distribution and growth are also affected by climate warming. In order to realize its economic value and ecological function, it is necessary to explore the impact of climate change on its suitable habitats under different scenarios. In this experiment, 181 geographical distribution data were collected, and the MaxEnt algorithm was used to predict the distribution of suitable habitats. To complete the simulation, we selected two greenhouse gas release scenarios, RCP4.5 and RCP8.5, and also three future time periods, 2025s, 2055s, and 2085s. The importance of environmental variables for modeling was evaluated by jackknife test. Our study found that accumulated temperature played a key role in the distribution of camphor trees. With the change of climate, the area of suitable range will increase and continue to move to the northwest of China. These findings could provide guidance for the plantation establishment and resource protection of camphor in China.

## 1. Introduction

Global climate is entering a stage of rapid change, which will bring enormous pressure on forest ecosystems during this century [1,2,3]. The consequent effects, such as temperature increase, precipitation variability and frequent droughts, are expected to have negative impacts on geographical distribution of trees and biodiversity of the plantations, and may lead to ecosystem change and species extinctions [4,5]. Many studies have reported that habitat suitability of trees is limited by global warming, such as *Argania spinosa* (L.) Skeels [6] and *Picea glauca* [7]. However, some reports predict that a moderate rise in temperature would be positive to the growth of tree species in temperate and some frigid regions [8,9,10,11]. Uncertainty of the impact of climate change on tree growth will inevitably lead to a change in suitable distribution area of tree species.

*Cinnamomum camphora* (L.) Presl is a precious multi-functional tree species [12,13], mainly distributed in the southern area of Yangtze River Basin in China [14]. The roots, stems, leaves, flowers and fruits of camphor are rich in natural camphor and ethereal oils, which are momentous industrial and pharmaceutical raw materials [15]. In addition, camphor plays a significant role in forest carbon sequestration, structure and function maintenance, and biodiversity conservation. Thus, exploring the impact of climate change on suitable habitats under different scenarios may be helpful to understand the superior economic value and ecological function of camphor [16].

In recent years, species distribution models have been widely used to assess the impacts of climate change on the distribution of species suitable areas [17,18,19]. These models can establish a connection between a species’ geographical distribution and environmental variables through statistical response functions [20,21,22], and then predict potential suitable areas and the future distribution of those species in the context of climate change over different times. While most species distribution models require collection of both presence and absence data of the geographical distribution, the Maximum Entropy Model (MaxEnt) needs only presence data (including species presence and environmental variables) [23]. MaxEnt can create the distribution map and variable response curve by testing the reserved part of the training data [24]. Some studies have shown that MaxEnt possesses preeminent predictive power in simulation and evaluation, and has been widely used in the study of distribution of current and future suitable habitats [16,25,26,27]. 

In this study, the MaxEnt algorithm was used to model the response of camphor to climate change, which will provide a theoretical basis for making management decisions and planting planning. Specifically, we tried to answer three questions: (1) What are the key climatic factors that affect the algorithm distribution of camphor? (2) Which habitats are suitable for camphor growth under current climatic conditions? (3) How will the future climate affect the habitat suitability of camphor?

## 2. Materials and Methods 

### 2.1. Data Collection

#### 2.1.1. Available Data 

In this study, we collected two types of data for modeling: one for the geographical location of camphor trees, which was used to describe the distribution of the species, and the other for climate data of the distribution areas, which described the climatic conditions of the habitats. Through field investigation, literature inquiry, and network resource search (http://www.cvh.ac.cn), we obtained 181 geographical distribution data of this species in China. This part of the data might show the autocorrelation in spatial distribution [28,29]. To alleviate this problem, we filtered the data at the spatial level. Specifically, we first meshed the study area, each grid with an area of 4 km × 4 km, with only a single distribution point selected in each grid. Furthermore, we required that the distribution points be at least 10 kilometers apart. The goal was to meet the terrain and environmental heterogeneity requirements of the model without excessively reducing the number of distribution points [30]. After filtering, 149 valid distribution data were finally obtained (Figure 1) and this data was used for MaxEnt modeling (Table A1). Among them, there were 43 field observations and 106 specimen records data. In addition, we collected the distribution data of 62 camphor trees from the global biodiversity information agency (GBIF) (1960–2018s, in China), including observation data and sample data. This part of data would be used to verify the division of suitable regions, and to evaluate the modeling effect.

#### 2.1.2. Climate Data

The accuracy of the plant climate response model depends mainly on whether the historical climate data of the plant habitat accurately reflects the actual climatic conditions [31]. Climate AP is a climate model used to represent the climate of the East Asia Pacific region, providing high-precision, high-resolution historical and future climate data [32]. It uses the best available climate data as baseline data and converts it to a scale-free format using dynamic local downscaling. It calculates and derives more biologically relevant climate variables that make it more useful in various applications in forest modeling. We used Climate AP to generate national current and future grid climate data with a spatial resolution of approximately 16 km^2^, which was used to predict the potential suitable range of camphor tree and its response to climate change. We used two greenhouse gas release scenarios, RCP4.5 and RCP8.5, and selected three future time periods, 2025s, 2055s, and 2085s, to represent the three periods of the early, middle, and final phases of the 21st century. The 16 climate variables in this study (Table 1) and the Climate AP client are available from the UBC server (http://asiapacific.forestry.ubc.ca/research-approaches/climate-modeling). To avoid cross-correlation within selected environmental variables, we used the Pearson correlation coefficient in the R language (version 3.5.1) for multicollinearity testing and eliminate variables with a correlation coefficient greater than 0.8 (Figure 2). The principal component analysis (PCA) was then used to select significant bioclimatic variables among the remaining variables [26]. Eventually, eight climate variables were retained for model building: mean annual temperature (MAT), temperature difference between mean warmest month temperature and mean coldest month temperature (TD), mean annual precipitation (MAP), degree days below 0 °C (DD < 0), precipitation as snow between August in previous year and July in current year (PAS), extreme minimum temperature over 30 years (EXT), Hargreaves climatic moisture deficit (CMD), and annual heat moisture index (AHM).

### 2.2. Model Establishment and Evaluation Methods

The MaxEnt method used in this study is based on the niche principle model, which assumes under certain known conditions, the system with the highest entropy is closest to its true state [33]. Firstly, the constraints are obtained according to the characteristics of the environmental variables of the species existence data, then the distribution probability of the maximum entropy is acquired under the constraint condition, and finally the habitat distribution of different spaces and time is predicted accordingly [23]. We chose the MaxEnt 3.4.1 software version and selected the following settings before the model runs: Checked the jackknife forecast to assess the importance of the variables in the model; random test = 25%; regularized multiplier = 1; maximum background points = 10,000; replicates = 10; the rest of the settings remained the default parameters. This setup is considered to be reasonable and effective in a wide range of niche studies [34].

To evaluate model performance, we randomly divided the data into training (75%) and validation data sets (25%). To consider the uncertainty introduced by training and validation set splitting, we generated 10 models using cross-validation methods with 10 iterations. The maximum, minimum and standard deviation obtained from repeated runs were used to estimate possible deviations due to arbitrary data splits, all of which were used for final prediction. Modeling results revealed the species–environment relationship and the contribution rate of each variable in the model. We used the jackknife test to investigate the importance of an individual climate variable for MaxEnt predictions and used the receiver operating characteristic curve (ROC) to assess the accuracy of the model. The area enclosed by the curve and the abscissa is the area under the curve (AUC) value, and the model performance range measured by the AUC is 0–1. The larger the value, the more the species distribution deviates from the random distribution. The evaluation criteria are as follows: AUC > 0.9 is very good; 0.8 < AUC < 0.9 is good; 0.7 < AUC < 0.8 is acceptable; 0.6 < AUC < 0.7 is bad; AUC < 0.6 is invalid [35]. The main advantage of this method is that it is independent of the threshold and the evaluation results are more objective [36].

In order to assist model validation and interpretation, it is usually necessary to set a decision threshold to distinguish the suitable and unsuitable regions. Threshold values could be set by many different approaches, in the cases of available absence data or presence only [37,38,39,40]. In this study, we applied a fixed threshold of 10%. Then, we projected the suitability of camphor distribution under current and future climatic conditions through MaxEnt, ranging from 0 (lowest suitability) to 1 (highest suitability). Based on the references published [25,41], we defined the estimated value of greater than 0.6 as a high-adapted region, 0.4–0.6 as a moderately-adapted region, 0.1–0.4 as a low-adapted region, and less than 0.1 as an unsuitable region. In terms of current and future climatic conditions under different emission scenarios and time periods, a suitable area map was generated for camphor.

## 3. Results

### 3.1. Model Performance and Evaluation

Results showed that the test data omission rate was very close to the prediction omission rate, and the fitting effect was good (Figure A1 in Appendix A). The average test AUC value of repeated operation was 0.923, and the standard deviation (SD) was 0.010, which indicated that the model is highly reliable for the potential habitat of camphor tree and can effectively reflect the distribution in China in the present and future time. The jackknife test was used during the MaxEnt model building to test the predictive power of each climate variable. Among the eight environmental variables used for model development, MAT, DD < 0, MAP, EXT and PAS performed well (AUC > 0.8, arranged from large to small), while AHM, TD, and CMD contributed less (Figure 2). The relative contribution of environmental variables to the MaxEnt model is shown in Table 1. The contribution rate of DD < 0 was higher, reaching 81.1%, followed by EXT (12.3%), and the contribution rate of residual variables was lower. According to the results of the jackknife test of variable importance, the environmental variable that decreased gain the most was EXT when it was omitted, which therefore appeared to have the most information that wasn’t present in the other variables (Figure A2 in Appendix A).

The response curve of habitat adaptability to environmental variables clarifies the quantitative relationship between the logistic probability and environmental variables. Moreover, we can deepen the understanding of the niche of this species by explaining the response of the eight variables to the adaptation, as shown in Figure 3. According to the response curves, the average annual temperature for suitable growth of camphor tree is 16.5–29 °C, the annual precipitation is 1000–3600 mm, and the extreme high temperature of 30 years is 37–39 °C. 

### 3.2. Potential Distribution Range

The ecological niche model predicted that climate warming would promote the expansion of the potential suitable habitats of camphor tree (Figure 4). In the current climate, highly suitable areas were concentrated in most areas south of the Yangtze River, accounting for 7.9% of land area in China. The prediction results were consistent with the survey records [42]. This study found that with the warming of the climate, the geographical distribution of high potential areas changed under the two scenarios, and the area of distribution range generally showed an increasing trend and gradually expanded to the high latitudes. Through horizontal observation, the predicted maps showed that Guizhou, Sichuan, Chongqing and Hubei would have a rise of probability presence. Longitudinal observation also showed that the range of suitable areas in Henan and Shandong would expand. Meanwhile, we found that there would be a significant reduction in the area of suitable habitats in Hainan, Guangdong and Guangxi coastal areas. In addition, it was predicted that by the end of this century, new suitable areas would appear in the northwestern part of Xinjiang Autonomous Region.

Comparing the two scenarios, we found that under the RCP8.5 condition, the area of suitable range would increase faster at high latitudes, but the area of suitable range at lower latitudes would reduce more (Figure 5). However, under the RCP4.5 scenario, the growth of the area of suitable habitat would gradually slow down, and would increase by only 4.7% from 2055s to 2085s. At the end of the century, under the RCP4.5 scenario, the area of suitable range would increase by 84.8% compared with the present, while the RCP8.5 scenario would increase by 1.3 times compared with the same period of the previous year. The predicted area of suitable range would account for 18.3% of land area in China.

## 4. Discussion

In this study, the ecological niche model was established using MaxEnt method to predict potential distribution areas of camphor tree on a national scale, and the model was considered to be reliable. Figure 4 depicts that 58 of the 62 sample points were distributed in the high-adaptive region, 2 in the middle-adaptive region and 2 in the low-adaptive region, respectively, and no distribution point was found in the non-adaptive region, showing that the predicted results were credible. We also found that among the temperature-related variables, DD < 0 was the most important contributor to the prediction of the suitable areas, which may be due to the fact that the accumulated temperature in cold months can seriously freeze the roots by lowering the soil temperature [43]. The results of the jackknife test of variable importance showed that EXT was very important in the simulation of camphor tree niche, indicating that extreme high temperature may have a serious inhibitory effect on the growth process. For example, the extreme high temperature increases breathing, reduces stomatal conductance, thereby reducing the need for higher transpiration, and to some extent directly reducing photosynthesis [44]. Our research showed that the average temperature of the most suitable growth areas of camphor tree is 16.5–29 °C, indicating sufficient heat accumulation is essential for the camphor growth. 

When used in isolation, the two precipitation-related variables of MAP and PAS were also important. The response curves showed that camphor tree had greater requirements for annual precipitation and was suitable for growing in areas with abundant rainfall. At the same time, areas with snowfall were extremely unsuitable for camphor growth. Some scholars have reported that factors such as soil moisture and soil nutrients, which were affected by precipitation, play an important role in tree growth [45]. However, compared with the relative variables contribution rates of niche models, MAP and PAS did not show a significant influence, indicating that the growth of camphor trees is more dependent on temperature variables. In the subtropical climate regions of southeastern China, the accumulated temperature is high and the precipitation is also abundant, making it an excellent habitat for camphor tree. However, under the trend of temperature rising, by the end of this century, the global temperature is expected to rise between 1.4 and 5.0 °C [8], which will lead to a serious decline of suitable areas in low latitudes such as Hainan. Camphor has poor cold resistance and is difficult to spread to high altitudes in the northwest [14], but the global warming trend will gradually break this restriction.

The model showed that under the background of climate change, the suitable range of camphor tree in China would gradually increase. Geographically, suitable areas would spread to the high latitudes in the northwest, while suitable areas in the lower latitudes of the southeast coast would decrease. These results were consistent with the speculation that the distribution of plants would move to high latitudes and high altitudes by global warming [46]. The expansion of potential suitable habitats will enhance competitiveness of camphor in ecological communities, which is a positive signal for afforestation and management. The concepts of entropy and mutual information index were developed by Shannon in the context of information theory [47]. They have been widely studied and applied in the case of multivariate normal distribution. An alternative way to define these climate variables that asses camphor distribution is using mutual information index to reduce the dimension of a multivariate system [48]. Considering the effects of climate variables on suitable areas of camphor by combining MaxEnt and mutual information index is a further work.

Modeling with MaxEnt inevitably encounters the problems of over-fitting and threshold selection. The selection of thresholds determines the prediction accuracy of potential distribution areas of species. Usually we need to pay attention to the difference between climate suitability and habitat suitability. In addition to climate, other characteristics such as soil type, matrix composition, slope and watershed are also important factors to affect the habitat suitability of species [49]. Due to the inadequate description of species environment requirements, the model may have the problem of being insufficient or over-fitting [50]. In addition, predicting species distribution needs to take into account their own physiological constraints and competition of ecological communities. Many species are currently geographically distributed to higher altitudes and latitudes, and the rate of movement has even exceeded the previously reported rate [51,52,53]. Our model predicted that the movement of suitable areas for camphor tree to the northwest was also consistent with the climate change rate of the general warming in northern China [54]. However, the rapid movement of species will inevitably affect the competition of bio-community populations, which may take decades or centuries for the richness and composition of the community to adapt to the current climate [55]. Moreover, changes in the actual distribution of species need to consider many factors, such as the delay of species response to external factors, individual physiological limitations, and driving changes, which may cause species distribution to lag behind climate change. The niche model established in this study emphasized the survival probability of this species under different climatic conditions. Therefore, we estimated that the actual natural migration rate of the boundary range will be lower than the prediction of the niche model.

Long-term climate change affects the regeneration and migration of ecosystem species, while short-term biological changes in local areas are dominated by non-climatic factors [56], which can be artificially controlled and interfered effectively. In the camphor tree afforestation activities, the highly suitable areas predicted by the model can be prioritized to achieve efficient use of resources. The emerging potential habitats in the northwestern part of Xinjiang are far apart and species migration is difficult. According to the species energy hypothesis, the cold regions that have experienced warming and the arid regions that have experienced increased water supply are expected to exhibit increased species richness [57]. Thus, we can also use them as choices for future afforestation areas. For existing plantations in the suitable areas, more attentions should be paid to management and protection. For example, the camphor tree is susceptible to pests, and climate warming increases the severity and frequency of pests and diseases [58]. Besides, parts of the current suitable areas may become unsuitable in the future, and the climatic conditions of the areas will not be conducive to the growth and development of this species. We should promptly formulate a reasonable contingency plan to help this species migrate to new suitable areas. Therefore, it is important to understand as much as possible the response of this species to climate fluctuations, and combine the niche model with the physiological characteristics to classify the suitable areas more scientifically.

## 5. Conclusions

In this study, we established the niche model of camphor tree by using MaxEnt algorithm, and found that accumulated temperature played a key role in the growth process of this species. In addition, this process was more dependent on temperature than precipitation conditions. At present, the potential habitats of camphor tree are mainly distributed in the subtropical regions of southeastern China. With climate change, the area of suitable range will expand and the habitats will continue to move to the northwest. We propose to incorporate the niche model into the development and conservation strategy of the camphor tree, which can guide the plantation establishment and resources protection of this species. 

## Figures and Tables

**Figure 1 ijerph-16-03185-f001:**
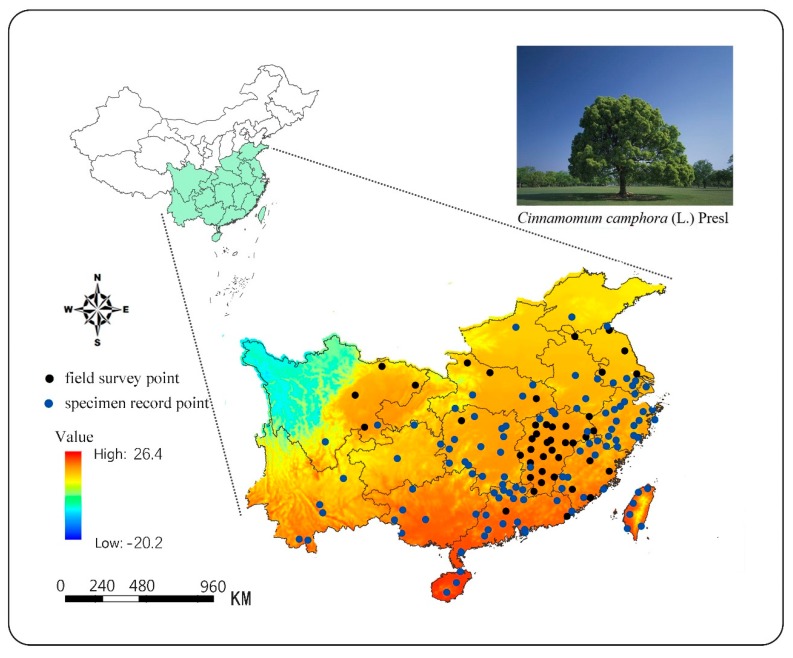
Distribution of the camphor presence points in major provinces with mean annual temperature.

**Figure 2 ijerph-16-03185-f002:**
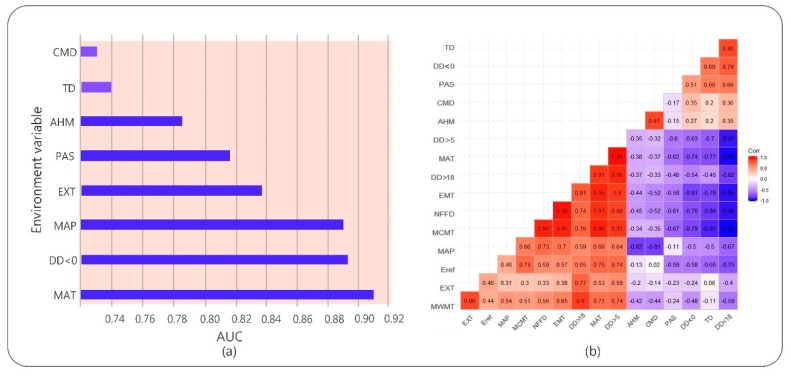
(**a**) Results of jackknife test for the area under the curve (AUC) of individual environmental variable importance relative to eight environmental variables for MaxEnt model. (**b**) Correlation analysis of the independent variables. The red squares are for positive correlations and the blue ones for negative correlations. The stronger the correlation, the darker the color.

**Figure 3 ijerph-16-03185-f003:**
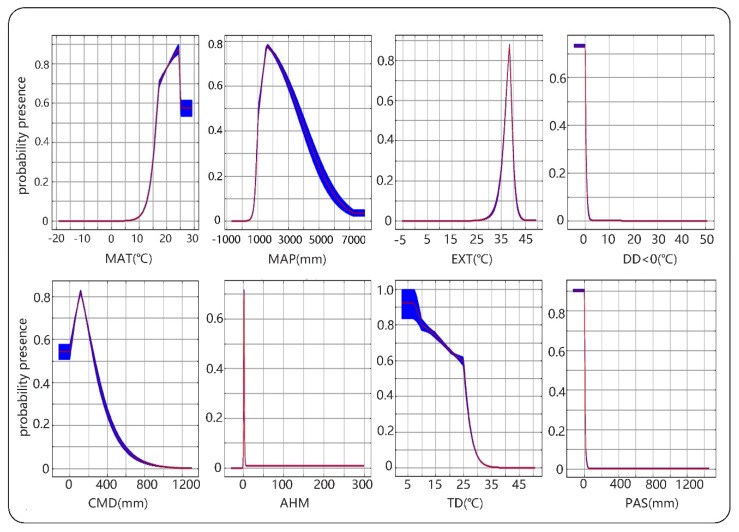
The response curves of eight environmental variables in camphor habitats distribution model. The logistic probability of presence is represented by the vertical axis and the climate variable by the horizontal axis. The values of probability presence range from 0 to 1, with values >0.5 meaning a better than random fit. The red curves shown are the averages over 10 replicate runs; blue margins show ±1 standard deviation (SD) calculated over 10 replicates.

**Figure 4 ijerph-16-03185-f004:**
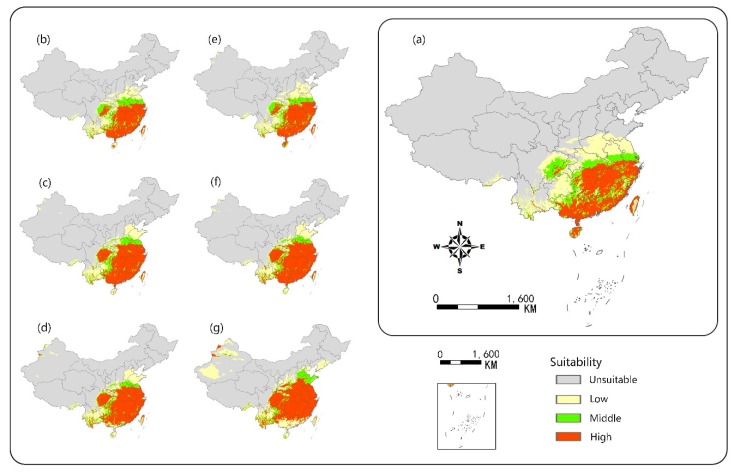
Potential habitat suitability of camphor tree projected by MaxEnt model. (**a**) Current occurrence; (**b**) based on RCP4.5 in 2025 s; (**c**) based on RCP4.5 in 2055 s; (**d**) based on RCP4.5 in 2085 s; (**e**) based on RCP8.5 in 2025 s; (**f**) based on RCP8.5 in 2055 s; (**g**) based on RCP8.5 in 2085 s.

**Figure 5 ijerph-16-03185-f005:**
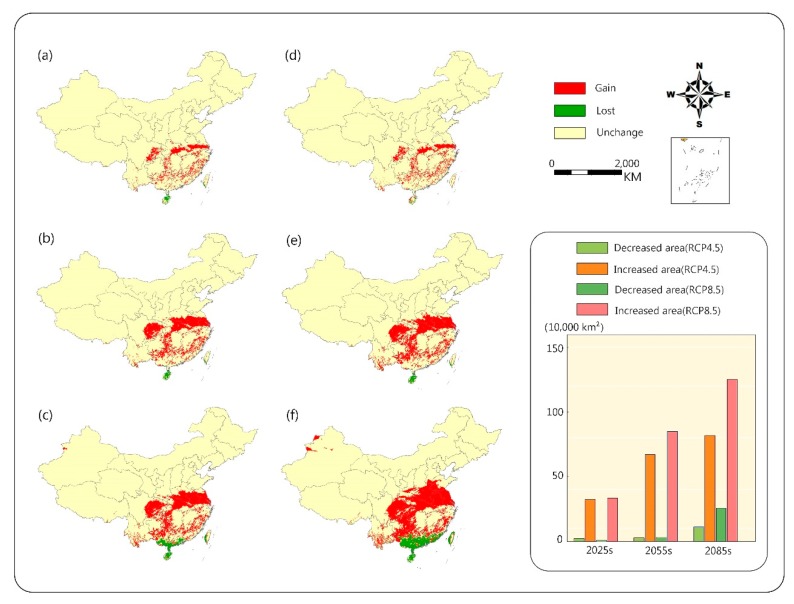
Change in habitat suitability of camphor tree in different periods in the future. RCP4.5: (**a**) 2025 s, (**b**) 2055 s, (**c**) 2085 s; RCP8.5: (**d**) 2025 s, (**e**) 2055 s, (**f**) 2085 s. The columnar graph shows the suitability changes in the area of different periods.

**Table 1 ijerph-16-03185-t001:** Climate variables provided by Climate AP and their percentage contribution. The variables with bold fonts were used in the model.

Code	Climate Variables	Units	% Contribution
**MAT**	**Mean annual temperature**	°C	2.0
MWMT	Mean warmest month temperature	°C	
MCMT	Mean coldest month temperature	°C	
**TD**	**Temperature difference between MWMT and MCMT, or continentality**	°C	0.6
**MAP**	**Mean annual precipitation**	mm	0.7
**EXT**	**Extreme maximum temperature over 30 years**	°C	12.3
**AHM**	**Annual heat moisture index (MAT+10)/(MAP/1000))**	**-**	0.0
DD > 5	Degree-days above 5 °C, growing degree-days	°C	
**DD < 0**	**Degree-days below 0 °C, chilling degree-days**	°C	81.1
NFFD	The number of frost-free days	day	
**PAS**	**Precipitation as snow between August in previous year and July in current year**	mm	0.2
EMT	Extreme minimum temperature over 30 years	°C	
Eref	Hargreaves reference evaporation	**-**	
**CMD**	**Hargreaves climatic moisture deficit**	**-**	3.0

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
