# Peer review of "Predictive Modeling of Suitable Habitats for Cinnamomum Camphora (L.) Presl Using Maxent Model under Climate Change in China"

_ijerph, 2019, doi:10.3390/ijerph16173185_

Round 1

Reviewer 1 Report

Predictive modeling of suitable habitats for Cinnamomum camphora (L.) Presl using MaxEnt model under climate change in China

By Zhang et al. (2019), submited to Forests ID: ijerph-568882

The authors consider the MaxEntr method to analize and predict Cinnamomum camphora distribution in some locations of China. As in Xhu et al. (2018, Forests), this method is very important to do these tasks, because MaxEnt allows to predict the spatial distribution of CC for several more decades, and provides several advantages over other methods, as is described by the authors.

Mayor comments.-

1. The paper presents some missing issues in MaxEnt method, explained in Section 2.2. I recommend to authors to following Xu et al. (2018) paper to give more (mathematical) details about MaxEnt. For example: how MaxEnt defined the contribution of each cimate variable? Xu et al. considered Principal component analysis (as a previous step of MaxEnt) to do this. Also, please provide more Jacknife test details.

2. On the another hand, I am according with authors about MaxEnt is a good method for prediction of tree distribution under several scenarios.

3. Conclusions section: An alternative way to define these factors (temp. and precip.) that asses Cinnamomum camphora (L.) Presl distribution is using Mutual Information Index (MII) (to reduce the dimension of a multivariate system), acordingly to Arellano-Valle et al. (2013), given that you consider MAxEnt method. Consider this comment for Discussion section as a further work.

Minor Comments.-

1. L24: "China".

2. L53: Also include the reference: Xu et al. (2018) about how MaxEnt method is applied for walnut (Juglans regia L.) species in China.

3. L66 & 83: "Presented" <-> "Available".

4. L105: "selected" <-> "used method".

5. L106: "thing" <-> "system".

6. L125: largest value?

7. L131: What type of reclassification is proccesed by Arcgis?

8. L137: Delete "Represent the".

Suggested References.-

1. Xu, X., Zhang, H., Yue, J., Xie, T., Xu, Y., & Tian, Y. (2018). Predicting shifts in the suitable climatic distribution of walnut (Juglans regia L.) in China: maximum entropy model paves the way to forest management. Forests, 9(3), 103.

2. Arellano-Valle, R.B., Contreras-Reyes, J.E., Genton, M.G. (2013). Shannon Entropy and Mutual Information for Multivariate Skew-Elliptical Distributions. Scandinavian Journal of Statistics, 40(1), 42-62. 

Reviewer 2 Report

The manuscript English language needs to be checked professionally or by the journal service. It suffers from improper choice of words and the sentences which need to be modified in many places. I have highlighted these is some pages but not fully as it very exhaustive to do so.

On the technical side, the text needs to be clearer. My main concern is about the classification (or re-classifying) of the predictions into few categories. This can potentially hugely affect the maps and the calculation of suitable areas. Although authors cited couple of papers for this purpose, it is not acceptable to arbitrary categorize the prediction maps. The threshold cut-off point which makes an area suitable or unsuitable can be defined based on MaxEnt output i.e. the table which gives the thresholds such as 10 percentile of training presence data. This depends on the aim of the study and sensitivity of the species studied. Please refer to the following papers to see how a map can be categorised or at least how to use the thresholds. It is possible to use one of the thresholds such as 10 percentile training presence logistic threshold and then arbitrarily categorize above that threshold.

The discussion part is jumping around and is hard to follow in many places because of the numerous language errors. At least in one paragraph the authors need to address the uncertainties or the caveats involved in using Species Distribution models such as MaxEnt. Please also refer to the following papers for this purpose.

There are also numerous grammatical comments in the attached pdf file.

For finding the threshold cut off point and caveats:

Narouei-Khandan, Hossein Ali. Ensemble models to assess the risk of exotic plant pathogens in a changing climate. Diss. Lincoln University, 2014.    Cited in Narouei-Khandan, H. A., et al. "Potential global and regional geographic distribution of Phomopsis vaccinii on Vaccinium species projected by two species distribution models." European Journal of Plant Pathology 148.4 (2017): 919-930.

Cao, Yong, et al. "Using Maxent to model the historic distributions of stonefly species in Illinois streams: the effects of regularization and threshold selections." Ecological Modelling 259 (2013): 30-39.

Padalia, Hitendra, Vivek Srivastava, and S. P. S. Kushwaha. "Modeling potential invasion range of alien invasive species, Hyptis suaveolens (L.) Poit. in India: Comparison of MaxEnt and GARP." Ecological informatics 22 (2014): 36-43.

Radosavljevic, Aleksandar, and Robert P. Anderson. "Making better Maxent models of species distributions: complexity, overfitting and evaluation." Journal of biogeography 41.4 (2014): 629-643.

For finding the proper threshold or cut off threshold

Young, Nick, Lane Carter, and Paul Evangelista. "A MaxEnt model v3. 3.3 e tutorial (ArcGIS v10)." Fort Collins, Colorado(2011).

Narouei-Khandan, Hossein A., et al. "Global climate suitability of citrus huanglongbing and its vector, the Asian citrus psyllid, using two correlative species distribution modeling approaches, with emphasis on the USA." European Journal of Plant Pathology 144.3 (2016): 655-670.

Round 2

Reviewer 1 Report

Thanks to author for considered my suggestion and comments. I can see only one mistake in reference [48], the correct authors list is: Arellano-Valle, R.B., Contreras-Reyes, J.E., Genton, M.G.

Author Response

Point 1: Thanks to author for considered my suggestion and comments. I can see only one mistake in reference [48], the correct authors list is: Arellano-Valle, R.B., Contreras-Reyes, J.E., Genton, M.G.

Response 1: According to your suggestion, we have corrected this mistake in reference [48]. Thank you very much.

Reviewer 2 Report

The authors have addressed the previous questions well. I have added some minor comments. The manuscript is not needed to be looked at by my anymore as long as the minor comments are addressed and a table of occurrence data is presented in the Appendix. 

Author Response

Point 1: The authors have addressed the previous questions well. I have added some minor comments. The manuscript is not needed to be looked at by my anymore as long as the minor comments are addressed and a table of occurrence data is presented in the Appendix.

Response 1: As you suggested, we revised the relevant comments and presented a table of occurrence data in the Appendix. Thank you very much.